# Adjustment of the Structure of the Simplest Amino Acid Present in Nature—Glycine, toward More Environmentally Friendly Ionic Forms of Phenoxypropionate-Based Herbicides

**DOI:** 10.3390/ijms24021360

**Published:** 2023-01-10

**Authors:** Adriana Olejniczak, Witold Stachowiak, Tomasz Rzemieniecki, Michał Niemczak

**Affiliations:** Department of Chemical Technology, Poznan University of Technology, 60-965 Poznan, Poland

**Keywords:** ionic liquids, 2-(2,4-dichlorophenoxy)propionate, glycine derivatives, sustainable agrochemistry, biological activity, AAILs

## Abstract

The use of chemicals for various purposes in agriculture has numerous consequences, such as the contamination of ecosystems. Thus, nowadays it is perceived that their development should adhere to the principles of green chemistry elaborated by Paul Anastas. Consequently, to create more environment-friendly herbicides, we elaborated a ‘green’ synthesis method of a series of ionic liquids (ILs) containing cations derived from glycine. The appropriately modified cations were combined with an anion from the group of phenoxy acids, commonly known as 2,4-DP. The products were obtained with high yields, and subsequently, their properties, such as density, viscosity and solubility, were thoroughly examined to elucidate existing structure–property relationships. All ILs were liquids at room temperature, which enabled the elimination of some serious issues associated with solid active forms, such as the polymorphism or precipitation of an active ingredient from spray solution. Additionally, the synthesized compounds were tested under greenhouse conditions, which allowed an assessment of their effectiveness in regulating the growth of oilseed rape, selected as a model dicotyledonous plant. The product comprising a dodecyl chain exhibited the greatest reduction in the fresh weight of plants, significantly surpassing not only a commercially used reference herbicide but also the potassium salt of 2,4-DP.

## 1. Introduction

The 12 principles of green chemistry introduced by Paul Anastas in 1998 are nowadays considered as the new pillars on which modern chemical syntheses should strongly rely [1,2,3]. Statistical data shown a gradual increase in financial investments focused on the implementation of the above-mentioned rules in various branches of industry. The market size of the global Green Chemicals Market in the year 2021 was valued at USD 9893.70 Million and is predicted to reach USD 16,684.27 Million by the year 2028. It is noteworthy that the majority of the reported ‘green’ initiatives are also directly related to a growing effort to slow down the progress of climate change and preserve existing ecosystems on our globe [4]. Therefore, the assessment of potential hazards caused by various chemicals including ILs has turned out to be extremely important, taking into account not only the large-scale activity, but also the laboratories in which the first concepts of the synthesis pathway are being developed [5].

The utilization of all renewable resources available on our planet turns out to be the most obvious, and yet the most detrimental solution, particularly for future generations that will inhabit the earth. The search for the most suitable raw materials for the synthesis of new compounds is a challenging task, not only for economic but also for logistic reasons [6]. However, natural amino acids can be an interesting source to obtain new chemical compounds [7]. Amino acids, as substances widely present in the environment, are supplied in the food chain or synthesized in living organisms, including humans. They can perform building or regenerative functions, but also participate in metabolic mechanisms [8]. The simplest representative of this group in terms of structure is glycine, which has a chemical formula NH_2_-CH_2_-COOH. Due to the presence of a nitrogen atom in its structure, it is easily possible to substitute a hydrogen atom with alkyl chains of various lengths. In effect, glycine could be an excellent candidate to become a source of cations in diverse variations of quaternary ammonium salts [9,10]. Moreover, the versatility of amino acids enables their utilization as a source of anion in the synthesis of ILs, which leads to novel bio-derived molecules exhibiting high potential in the fields of functional food, drug delivery systems and production of biodegradable plastics [11]. Derivatization of amino acids [12] including glycine is the subject of the research of many scientists from various scientific facilities, wherein among others, surface, biological or phytotoxic properties of new compounds are being thoroughly examined. It has been demonstrated that ammonium salts based on glycine as the ion of natural origin are characterized by excellent surface properties or the ability to regulate plant growth, which can potentially be utilized in the development of novel and environmentally friendly forms of various agrochemicals [13,14,15,16].

Among the family of phenoxy acids used in commercial preparations focused on the eradication of unwanted vegetation, 2-(2,4-dichlorophenoxy)propionic acid, also commercially known as 2,4-DP or dichloroprop, deserves special attention [17]. This particular compound is still widely used to control the growth of broad-leaved aquatic weeds as well as annual and perennial weeds in cereals or pastures [18]. In addition, current reports have confirmed its positive effect on fruit growth, e.g., in the case of Valencia oranges (*Citrus sinensis* Osb.) the fruits were characterized by increased weight, which resulted in an increased yield per unit of crop area [19]. The susceptibility to degradation of 2,4-DP by numerous bacterial strains, such as *Sphingomonas herbicidovorans MH*, *Rhodoferax* sp. *P230* and *Delftia acidovorans MC1* should be perceived as an additional advantage, particularly in terms of modern formulations in agriculture, designed not only to be effective but also to be harmless to the biosphere [20].

The current state-of-the-art reports only 2,4-DP-based ILs comprising fully synthetic ammonium cations substituted with straight alkyl or alkylaryl chains that have been recognized as potentially hazardous, mainly due to poor biodegradation as well as the toxic effect to aquatic life or mammals after ingestion [21,22,23,24]. Therefore, derivatization of amino acids of natural origin into compatible eco-friendly cations that can be combined with the herbicidal anion mentioned above can be considered as the next step in the development of novel active ingredients with a potentially low environmental impact. As a consequence, in the present study new alkylated analogues of glycine were successfully combined with 2,4-DP with high yields via a three-step procedure that used the synergy of the principles of green chemistry [3]. A series of the obtained novel ILs were evaluated in terms of their physicochemical properties, such as density, viscosity or solubility in various solvents that are frequently required in the process of registration of new agrochemicals on the market. Additionally, such parameters provide valuable information regarding the nanostructure of the formed ionic pairs (including hydrogen bonding network, strength of coulombic and van der Waals interactions, energy of crystal lattice, tendency to form aggregates), that are crucial in terms of designing other ILs exhibiting desired functions [25].

## 2. Results and Discussion

### 2.1. Synthesis of Alkyldimethylglycine Hydrochlorides

Synthesis of ILs containing 2-(2,4-dichlorophenoxy)propionate anion and alkyldimethylglycinium cations consisted of three main stages. Initially, potassium chloroacetate was obtained according to the methodology given in Section 3.2. The product was a white solid at room temperature with a melting point in a range 275–278 °C. The first stage of the synthesis involved the quaternization of the appropriate alkyldimethylamine with the prepared potassium chloroacetate (Figure 1). In accordance with the methodology included in point 2.3, quaternization reactions were carried out in methanol, wherein the by-product—potassium chloride precipitated from the reaction mixture. Separation of the precipitate and the subsequent evaporation of methanol from the filtrate allowed seven alkyldimethylglycines to be obtained in the zwitterionic form.

The concentrations of both the alkyldimethylamine and zwitterionic product in the reactor could be evaluated on the basis of the intensity of signals in the FT-IR spectra collected in situ throughout the entire first stage of the process. Thus, tracking the changes of the relative absorbance of two characteristic bands: 1249 cm^−1^ (C–N stretch, alkyldimethylamine) and 1639 cm^−1^ (C=O stretch, zwitterionic alkyldimethylglycine) allowed the time after which the conversion rate of a substrate was favorable to be determined. Analyses were performed for the quaternization of two selected substrates comprising different lengths of alkyl chains: butyldimethylamine and dodecyldimethylamine (Figure 1). In the case of butyldimethylamine, a conversion rate of 98% was achieved after 14 h. On the contrary, the quaternization of dodecyldimethylamine was characterized by a notably lower reaction rate and the equilibrium state was reached after 42 h. Nevertheless, a conversion rate of dodecyldimethylamine as high as 95% was achieved after 24 h. Generally, all efforts focused on the further extension of the process yield are often considered as uneconomical; therefore, 24 h was established as the favorable reaction time for this stage.

Nonetheless, according to previous findings, tertiary aliphatic amines can be effectively quaternized in much less time, even within 4 h [2,26]. Moreover, structurally more complex compounds with an amine group, such as quinine, can be analogously converted to their quaternary forms using bromoethane, whereas the reaction time varies from 10 to 15 h [27]. With respect to the above reports, we believe that the considerably slower formation of zwitterionic alkyldimethylglycines was associated with the use of methanol as reaction medium. It is widely known that methanol as a protic solvent promotes the quaternization reaction (occurring according to the S_N_2 mechanism) to a lesser extent compared to aprotic solvents, such as acetonitrile. However, the choice of methanol was dictated by its ability to dissolve potassium chloroacetate, which, in this particular case, greatly facilitated the reaction of the quaternizing agent with the amine. Interestingly, in the case of more reactive alkylating agents (e.g., alkyl chloromethyl ethers that react according to the S_N_1 mechanism), the methanolic environment additionally promotes the reaction. Therefore, this solvent is usually replaced by compounds exhibiting much lower polarity, such as hexane, to ensure all necessary safety measures [28].

The second stage of the process involved a protonation reaction of the obtained quaternized products, wherein hydrochloric acid was utilized as a proton donor (Figure 2). According to the methodology provided in Section 3.4., the alkyldimethylglycine hydrochlorides, precipitated from acetone, were filtered off and dried.

Ultimately, seven alkyldimethylglycine hydrochlorides, which comprised alkyl chains of various lengths—from butyl (**1**) to hexadecyl (**7**) were obtained via a sustainable and an environmentally friendly method. The synthesized products are described in Table 1. The total yield of steps I and II was high and ranged from 70% for **2** to 93% for **7**. All compounds (**1**–**7**) were found to be white solids at room temperature, in contrast to alkyldimethylglycines in a zwitterionic form, which were highly viscous greases. The melting points of **1**–**7** ranged from 119 °C for the product with butyl substituent (**1**) to 160 °C for compounds with the octyl (**3**) and dodecyl (**5**) group.

The structures of the selected products (**3**–**7**) were confirmed by proton and carbon nuclear resonance (^1^H and ^13^C NMR). Appendix A, summarize the chemical shifts of selected protons in the ^1^H and ^13^C NMR spectra of the obtained hydrochlorides. In the ^1^H NMR spectra, we can distinguish the most characteristic peaks, such as the singlet at 11.00–11.03 ppm (originating from the proton present in the carboxyl group), the singlet at 4.32–4.42 ppm (attributed to two protons in the methylene group), the singlet at 3.22–3.31 ppm (from protons present in two methyl groups attached to the nitrogen atom) and the triplet at 0.86–0.90 ppm (originating from three protons attached to the last carbon in the alkyl substituent). The analysis of the ^13^C NMR spectra revealed the following characteristic shifts: 166.7–167.7 ppm (from carbon in carboxyl group), 60.5–61.9 ppm (from carbon in methylene group), 50.5–52.1 ppm (from two methyl groups attached to nitrogen) and 13.9–14.5 ppm (attributed to the last carbon in the alkyl group).

### 2.2. Synthesis of Alkyldimethylglycinium 2-(2,4-Dichlorophenoxy)propionates

In the third step, alkyldimethylglycinium 2-(2,4-dichlorophenoxy)propionates were synthesized according to Figure 3. All reactions were carried out according to the methodology included in point 2.5. The appropriate hydrochloride was added to the reaction vessel containing a potassium salt of herbicide to initiate the anion exchange reaction. After purification, a homologous series consisting of seven alkyldimethylglycinium 2-(2,4-dichlorophenoxy)propionates (**8**–**14**) was obtained.

The yields of the anion exchange reactions, conducted via a synthesis path that adhered to the principles of green chemistry, were high and generally exceeded 90% (Table 2). To emphasize the sustainable and eco-friendly nature of the reported novel chemicals as safe and simultaneously economical alternatives for common plant protection products, calculations were made in order to assess their Green Chemistry Metrics [29] (see Appendix A). At room temperature all the obtained products (**8**–**14**) were highly viscous pale-yellow liquids. Nonetheless, due to a melting point below 100 °C they can be classified as new ILs. It should be stressed that the liquefaction of active ingredients in agrochemistry can be particularly beneficial as it eliminates the problems associated with solid forms of substances (e.g., spontaneous crystallization, polymorphism, low solubility or bioavailability). Therefore, the development of novel active ingredients that exist in liquid form complies not only with the concept of green chemistry, but also with the rules of sustainable agriculture. Since the obtained products were found to be highly hygroscopic, despite storage in a vacuum desiccator over P_4_O_10_ the amount of water in ILs was considerable and amounted to approx. 0.9–1.6%.

The structures of alkyldimethylglycinium 2-(2,4-dichlorophenoxy)propionates (**8**, **9**, **11**, **12**, **14**) were confirmed by proton and carbon nuclear resonance. Appendix A, summarize the values of the chemical shifts in the collected ^1^H and ^13^C NMR spectra. In ^1^H NMR spectra of the obtained ILs, signals originating from the cation were observed in analogous locations as in the case of hydrochlorides **1**–**7**. Additionally, the presence of an anion was confirmed by one doublet at 1.62–1.64 ppm (three protons from the methyl substituent), one quartet at 4.64–4.65 ppm (protons from the methylene group) and three doublets at 6.83–6.84, 7.10–7.11 and 7.32–7.33 ppm (protons attached to the aromatic ring). Furthermore, the presence of an anion in the ^13^C NMR spectra was confirmed by the following of signals: 174.1–174.6 ppm (from the carboxylate group), 152.4–152.6, 129.6–129.7 and 127.3–127.4 ppm (from the benzene ring), 74.3–74.7 ppm (from the carbon in the methylene group) and 18.3–18.4 ppm (carbon from the methyl substituent).

### 2.3. Physicochemical Properties of the Obtained Products

#### 2.3.1. Solubility

The solubility at 25 °C of alkyldimethylglycine hydrochlorides (**1**–**7**) and their derivatives containing a herbicidal anion (**8**–**14**) was analyzed according to the methodology described in Section 3.8. The results, shown in Table 3, indicate that all the hydrochlorides tested were soluble in methanol. Moreover, the majority of them dissolved in water and DMSO as well. However, as the length of the alkyl chain in the cation was increasing, a reduction in the solubility in both solvents was noted. All hydrochlorides were found to be insoluble in medium polarity solvents such as acetone, acetonitrile and ethyl acetate. Nonetheless, salts comprising medium-length alkyl chains such as octyl (**3**), decyl (**4**) and dodecyl (**5**) exhibited a partial affinity for 2-propanol. None of the obtained hydrochlorides dissolved in the least polar solvents such as hexane and toluene, which was mainly due to the presence of highly polar ionic bonds in their structures [15].

Interestingly, replacement of the chloride anion for 2-(2,4-dichlorophenoxy)propionate significantly improved the solubility of the obtained products (**8**–**14**) in organic solvents. All ILs turned out to be easily or moderately soluble in acetone, acetonitrile, methanol, DMSO, ethyl acetate, chloroform and toluene. However, only three compounds that contained the longest alkyl substituents in the cation, such as dodecyl (**12**), tetradecyl (**13**), and hexadecyl (**14**) exhibited moderate solubility in hexane. The insertion of herbicidal anion into ILs’ structure increased their hydrophobicity. It can be considered as a beneficial effect due to the fact that good lipophilicity could improve the penetration of the leaf surface by the active ingredients, allowing the target tissues to be reached more efficiently, consequently improving herbicidal activity [30,31,32].

#### 2.3.2. Density

The values of density at the temperature range from 20 to 80 °C were determined for products that were liquids at room temperature–ILs **8**–**14** (see Appendix A). Before the measurements (described in the Section 3.9), compounds were stored in a vacuum desiccator over P_4_O_10_. The results presented in Figure 2 (as well as in Appendix A) indicate that the density of ILs was decreasing with the increase in the length of the alkyl chain. Therefore, at 20 °C the density of all tested ILs ranged from 1.07 g cm^−3^ for **14** to 1.22 g cm^−3^ for **8**. This trend can be explained by the increase in the molar volumes of ILs’ molecules caused by the elongation of the alkyl chain [33].

In the case of all analyzed ILs, the density values decreased linearly with the increase in the temperature, which was found to be consistent with the available data in the literature [33,34]. Eventually, at the temperature of 80 °C, the density of products ranged from 1.03 cm^−3^ for **14** to 1.18 g cm^−3^ for **8**. Additionally, the experimental density (*ρ*) values were approximated to a straight line via linear regression according to the following empirical Equation (1):ln *ρ* = −λ·T + *ρ*_0_
(1)
where *ρ*_0_ is an empirical constant–standard density (at 0 °C) and λ is a thermal expansion coefficient. 

Moreover, on the basis of experimental density values, the molecular volume (V) at T = 298.15 K, the lattice energy (UPOT) and the standard molar entropy (Sº) at T = 298.15 K were calculated using the following equations [35]:V = M_w_⁄(N·*ρ*)(2)
U_POT_ = 1981.2·[(*ρ*⁄M_w_)^0.33^] + 103.8(3)
Sº = 1246.5 (V) + 29.5(4)
where M_w_ is the molar mass and N is Avogadro’s constant (6.022·10^23^ mol^−1^). The following parameters λ, *ρ*_0_, V, Sº, U_POT_, calculated from the equations listed above are in Table 4.

The values of λ for all ILs occurred in the range from 6.6·10^−4^ to 7.7·10^−4^, which is lower than those of common solvents, such as short alkyl alcohols [36]. Due to the increase in the alkyl length, the molecular volume of ILs increased from approx. 0.533 nm^3^ for **8** to 0.867 nm^3^ for **14**. The lattice potential energy calculated for ILs decreased with the increasing carbon chain length of in the cation. These data are in agreement with other reports [37], wherein this phenomenon was explained by the fact that alkyl chain elongation generally results in an increase in the entropy of compounds. In the case of the obtained products this parameter ranged from approx. 694 for **8** to 1110 J·K^−1^·mol^−1^ for **14**. As a consequence, the packing efficiency in the synthesized ILs with longer alkyls was substantially reduced.

#### 2.3.3. Refractive Index 

The values of the refractive index were measured for ILs **8**–**14** (see Appendix A). Description of the analysis is provided in Section 3.10 and the results are shown in Figure 3. It should be noted that the refractive index at 20 °C (n_D_^20^) is a characteristic value for each chemical compound; therefore, the experimental n_D_^20^ values for **8**–**14** are additionally summarized in Table 4. At 20 °C, this parameter was highest (1.522) for the product with the shortest alkyl substituent (**8**), while the lowest value (1.497) was recorded for IL with the tetradecyl substituent (**13**) (see Appendix A). Additionally, the increase in temperature caused a linear decrease in the refractive index in the case of all compounds (Figure 3), which is consistent with the data in the literature [33,34,38]. The linear regression allowed equations *y = a·x + b* to be determined for each compound, where *a* amounted to approximately 3·10^−4^, while *b* ranged from 1.501 for **13** to 1.529 for **8**. Finally, at 80 °C the values of the refractive index ranged from 1.479 (for **14**) to 1.499 (for **8**).

It is known that the refractive index of ILs generally increases with increasing density. The refractive index of a compound is higher when its molecules are more tightly packed, which is directly associated with a higher density [39]. Taking into account this concept, we noticed that the refractive indices of products **8**–**14** were correlated with the values of density described in the previous section. Therefore, ILs characterized by the lowest values of refractive index (**13** and **14**) possessed the lowest density as well. The following study proved that the ‘density–refractive index’ relationship is also valid in the case of homologous series of ILs comprising derivatives of glycine as the cations.

#### 2.3.4. Viscosity

Viscosity analysis was performed at the temperature range from 20 to 80 °C for ILs **8**–**14** in accordance with the methodology included in Section 3.11. (see Appendix A). It was established that the tested compounds were Newtonian fluids, since for each IL the shear stress was proportional to the shear rate value (see Appendix A). At 20 °C, the highest viscosities (see Appendix A) were observed for ILs with short alkyl chains (**9** and **10**), which amounted to 1720 Pa·s and 1800 Pa·s, respectively. Values for the other ILs were notably lower and ranged from approximately 5 to 480 Pa·s. In Figure 4 one can notice a sharp decrease in viscosity in the temperature range from 20 to 40 °C. However, a further increase in temperature had a much lower impact on the reduction in this parameter. Finally, at 80 °C, viscosities ranged from 0.15 Pa·s for **13** to 1.01 Pa·s for **3**.

It should be emphasized that the obtained results were much higher than the values determined for other ILs. This was most likely due to the presence of an ionic bond in the cation and carboxyl group, which is capable of forming intermolecular hydrogen bond network [40]. Moreover, according to the literature, the viscosity of ILs generally increases with the elongation of the alkyl chain in the cation [41,42,43,44]. Different results for the obtained ILs (see Appendix A) may indicate that the presence of long alkyl chains increased the distance between individual ions, which significantly hindered the formation of hydrogen bonds that are often responsible for an increase in viscosity values.

Generally, the temperature dependence of the viscosity for ILs approximately follows the Arrhenius Equation (5):ln η = ln η_∞_ + E_a_/(R·T)(5)
where η is the dynamic viscosity, E_a_ is the activation energy for viscous flows (and represents the energy barrier that needs to be overcome by ion or mass transport), R is the universal gas constant (8.314 J·mol^−1^·K^−1^), T is the measurement temperature and ln η_∞_ is the natural logarithm of the viscosity at infinite temperature.

As presented in Appendix A, the investigated ILs exhibited a linear relationship which allowed us to determine the equation fitting parameters via a least-squares method. The high R^2^ values for all ILs, greater than 0.90, indicate that all ILs were well fitted with the proposed Arrhenius model. The calculated E_a_ and ln η_∞_ values varied from 61.1 kJ mol^−1^ for **13** to 109.4 kJ mol^−1^ for **9** and from −37.8 kJ mol^−1^ for **9** to −24.2 kJ mol^−1^ for **13**, respectively (see Table 4). According to the literature, the majority of the ILs are characterized by E_a_ values of 20–50 kJ mol^−1^ [34,45]. However, even the 3-fold larger E_a_ values for the synthesized ILs show that their viscosity was more responsive to a temperature change.

### 2.4. Biological Activity

The efficacy of a synthesized homologous series of 2,4-DP-based ILs (**8**–**14**) has been determined in greenhouse experiments, wherein oilseed rape was selected as a model dicotyledonous plant (exact values with statistical analysis are provided in Appendix A). The collected data shown in Figure 5 clearly indicate that a combination of a herbicidal anion with derivatives of glycine did not result in loss of biological activity. Bearing in mind that some amino acids, such as glycine, are commonly applied as plant growth stimulators or substances that reduce the effect of stress, the obtained results bring scientifically important knowledge on the effect of a combination of two ions exhibiting a potentially antagonistic effect [46,47,48]. Interestingly, the phytotoxic properties increased with the elongation of the alkyl chain attached to the nitrogen atom, reaching the highest value (55% reduction in fresh weight) in the case of compound comprising a dodecyl substituent (**12**). However, further elongation of the alkyl chain led to a decrease in biological activity, which could be attributed to the so-called ‘cut-off’ effect. As a result of hydrophobicity that was too great, the bioavailability of the active ingredient or its translocation in the plant can be significantly reduced, which was also noticed in a few of the previous reports describing ILs containing other herbicidally active anions [13,49].

Within the group of synthesized products, IL **8** comprising the shortest alkyl (butyl) showed the lowest phytotoxicity, reducing the growth of the tested plants by approximately 17% (which was a 3-fold worse result compared to the most effective compound (**12**)). These values are consistent with available reports, according to which a sufficient level of amphipilicity is required to ensure ILs’ good adsorption into the plants through a hydrophobic surface of the leaf [13,50]. The following assumption is also consistent with results regarding the potassium salt of 2,4-DP that turned out to be lower in comparison to some amphiphilic ILs, particularly products **12** and **13**. It should also be emphasized that one of the reference substances—commercial preparation available on the market (Aminopielik Standard 600 SL) showed much weaker phytotoxic activity (11%) toward oilseed rape compared to that of all tested ILs. Therefore, the described results allow us to conclude that there is no need to implement potential activity enhancers (so called adjuvants) in further possible formulations comprising these ILs.

In light of recent reports demonstrating an extremely detrimental influence of some of these additives (particularly ethoxylated tallow amines) on human health and ecosystem, their elimination from developed formulations can bring substantial advantages in terms of protection of living organisms at various trophic levels [51,52,53].

## 3. Materials and Methods

### 3.1. Materials

Butyldimethylamine 99%, hexyldimethylamine 98%, octyldimethylamine 95%, decyldimethylamine 98%, dodecyldimethylamine 97%, tetradecyldimethylamine 95%, hexadecyldimethylamine 95% were purchased from Sigma-Aldrich (St. Louis, MO, USA). 2-(2,4-dichlorophenoxy)propionic acid 98% and chloroacetic acid 99% were obtained from Alfa Aesar (Haverhill, MA, USA). All solvents (methanol, DMSO, acetonitrile, acetone, 2-propanol, ethyl acetate, chloroform, toluene, hexane) and potassium hydroxide were delivered by Avantor (Gliwice, Poland) and used without further purification. Deionized water with a conductivity <0.1 μS·cm^−1^, was used from the Hydrolab HLP Smart 1000 demineralizer (Straszyn, Poland).

### 3.2. Synthesis of Potassium Chloroacetate

A total of 1.59 mol of chloroacetic acid was introduced into a 1000 cm^3^ round bottom flask and dissolved in 200 cm^3^ of methanol. Then, a stoichiometric amount of potassium hydroxide with a purity of 85%, dissolved in 150 cm^3^ of methanol, was added in small portions with constant stirring. The conversion rate of chloroacetic acid was monitored with an S47 Seven Multi pH-meter (Mettler Toledo, Columbus, OH, USA). After complete neutralization, the post-reaction mixture was cooled to −20 °C. Next, the formed precipitate of potassium chloroacetate was filtered off, washed three times with cooled methanol and dried under vacuum for 48 h at 50 °C. The yield of the synthesis amounted to approximately 99%.

### 3.3. Synthesis of Alkyldimethylglycine Zwitterions

A total of 0.10 mol of an appropriate alkyldimethylamine was introduced into the round-bottom flask and dissolved in 50 cm^3^ of methanol. Then, a suspension containing 0.10 mol of potassium chloroacetate and 50 cm^3^ of methanol was added. The mixture was stirred at 60 °C under reflux for 24 h. The course of the reaction was monitored in a semi-automated EasyMax 102 reactor (Mettler Toledo) connected with a ReactIR iC15 probe equipped with an MCT detector and a 9.5-mm AgX probe with a diamond tip. Subsequently, the precipitated by-product (potassium chloride) was filtered off and the methanol was evaporated with the use of the rotary evaporator. The yields of the obtained products varied from 94 to 96%.

### 3.4. Synthesis of Alkyldimethylglycine Hydrochlorides

A total of 0.30 mol of an appropriate alkyldimethylglycine in zwitterionic form was introduced into the round-bottom flask. Then, 50 cm^3^ of acetone and a stoichiometric amount (0.30 mol) of 35% aqueous solution of hydrochloric acid was added to each flask containing the appropriate alkyldimethylglycine in zwitterionic form. The mixture was stirred for 15 min at an ambient temperature to maximize the efficiency of the protonation reaction. Then the post reaction mixture was cooled to −20 °C to initiate the precipitation of the product. Next, the precipitate (alkyldimethylglycine hydrochloride) was filtered off, washed three times with cold acetone (−20 °C) and dried under vacuum at 50 °C for 24 h. The yields of the synthesized alkyldimethylglycine hydrochlorides ranged from 74 to 98%.

### 3.5. Synthesis of Dimethylglycinium 2-(2,4-Dichlorophenoxy)propionates

A total of 0.0153 mol of 2-(2,4-dichlorophenoxy)propionic acid was placed in a 100 cm^3^ round bottom flask and dissolved in 10 cm^3^ of methanol. Then, an equimolar amount of potassium hydroxide (0.0153 mol) dissolved in 10 cm^3^ of methanol was added in order to obtain potassium 2-(2,4-dichlorophenoxy)propionate. The course of the neutralization reaction was monitored in a semi-automated EasyMax 102 reactor (Mettler Toledo) equipped with a S47 Seven Multi pH meter (Mettler Toledo). Then, 0.0150 mol of the appropriate alkyldimethylamine hydrochloride was introduced into the reaction system. The mixture was vigorously stirred at ambient temperature for 3 h to ensure the complete ion exchange reaction. Then, the solution was cooled to −20 °C and the precipitated inorganic salt by-product was filtered off. Next, the methanol was evaporated from the filtrate with the use of a rotary evaporator. In order to purify the product from the residual inorganic salt as well as the excess of used substrate, each compound was dissolved in 30 cm^3^ of acetone. Then, the formed precipitate was filtered off and the solvent was evaporated on a rotary evaporator. Finally, the product was dried under reduced pressure at 50 °C for 24 h. The yields of the anion exchange reactions occurred within the range of 90–96%.

### 3.6. Spectral Analysis

^1^H NMR spectra were recorded on a Mercury Gemini 300 spectrometer operating at 300 MHz spectrometer with TMS as the internal standard. ^13^C NMR spectra were obtained with the same instruments at 75 MHz. Deuterated methanol, chloroform and dimethyl sulfoxide (DMSO) were used as solvents for analyses.

### 3.7. Melting Point

The melting points of the obtained salts were measured using a Melting Point B-540 (Büchi, Bretten, Germany) apparatus. The device was performed using reference materials with defined melting points (vanillin—81.7 °C, phenacetin—134.8 °C and caffeine—234.0 °C). The heating rate of the tested samples was equal to 5 °C/min. The melting point was visually determined for each sample.

### 3.8. Solubility

The solubility of the prepared ILs in ten representative solvents was determined according to the protocols in Vogel’s Textbook of Practical Organic Chemistry [54]. The solvents chosen for study were arranged in order of the descending value of their Snyder polarity index [55]: water—9.0, methanol—6.6, DMSO—6.5, acetonitrile—6.2, acetone—5.1, ethyl acetate—4.3, 2-propanol—4.3, chloroform—4.1, toluene—2.3 and hexane—0.0. A 0.1 g sample of each IL was added to a certain volume of solvent and the samples were thermostated in water bath WNB Model 7 (MEMMERT, Schwabach, Germany) at 25 °C. Based on the volume of solvent used, 3 types of behaviors were recorded: ‘good solubility’ applied to compounds which dissolved in 1 cm^3^ of solvent (>10%), ‘medium solubility’—applied to compounds that dissolved in 3 cm^3^ of solvent (3.3–10%), and ‘low solubility’—applied to the compounds which did not dissolve in 3 cm^3^ of solvent (<3.3%).

### 3.9. Density

Density was determined by using an Automatic Density Meter DDM2911 (Rudolph Research Analytical, Hackettstown, NJ, USA) with the mechanical oscillator method. The density of ILs (about 1.0 cm^−3^) was measured in the range from 20 to 80 °C, and the temperature was controlled with a Peltier module. The accuracy of the temperature stabilization was equal to 0.02 °C. Before the series of measurements, the apparatus was subjected to a two-point calibration using deionized water and air as the references. The test was carried out in accordance with the following methodology: The U-tube in the device was filled with the tested liquid in the amount of 1.0 to 1.5 cm^3^. The analysis was performed by measuring the frequency of the U-tube vibrations, which depended on the density of the tested compound. After each series of measurements, the densimeter was washed with water and organic solvents (methanol and acetone) and dried with airflow.

### 3.10. Refractive Index

The refractive index was specified using an Automatic Refractometer J357 (Rudolph Research Analytical) with an electronic temperature control. Approximately 1 cm^3^ of tested compounds was analyzed at a temperature from 20 °C to 80 °C with a 10 °C step. The calibration of the apparatus was performed using water as the reference material. The uncertainty of the measurement was less than 0.00005.

### 3.11. Viscosity

Viscosity was determined using a RC30-CPS rheometer (RheoTec Messtechnik GmbH, Dresden, Germany) with cone-shaped geometry (C50-2 with the range of measured viscosities within the range of 0.005 to 5000 Pa·s). Liquid samples with a volume of approximately 1.5 cm^3^ were used for the study. The viscosity was measured in the temperature range from 20 to 80 °C. The uncertainty of the viscosity measurement was estimated to be less than 10^−4^ Pa·s.

### 3.12. Biological Activity

The oilseed rape (*Brassica napus* L.) plants were grown in 0.5 L plastic pots containing commercial peat-based potting material. Within 10 days after emergence, the plants were thinned to five per pot and watered as needed. All ILs were dissolved in water in a dose of 400 g of (*R*)-2-(2,4-dichlorophenoxy)propionate anion per 1 ha (which corresponded to 800 g of the obtained products containing a racemic mixture of 2,4-DP per 1 ha). Treatments were applied at the four-leaf stage using a moving sprayer occupied with Tee Jet 110/02 flat-fan nozzles delivering 200 L of spray solution per 1 ha at 0.2 MPa pressure. The plants were placed in a greenhouse at a temperature of 20 °C, a humidity of 60% and a photoperiod of 16/8 (day/night hours). The study was carried out in four replications in a completely randomized setup. Potassium 2-(2,4-dichlorophenoxy)propionate and Aminopielik Standard 600 SL (containing dimethylammonium salt of 2,4-dichlorophenoxyacetic acid) were utilized as reference herbicides at the same dose of active ingredient. After 2 weeks, the plants were then cut to soil level and weighed (with 0.01 g accuracy). The reduction in the fresh weight was determined in comparison to the treatment with pure water (control).

To properly assess the significant differences in the determined phytotoxic effects, a statistical analysis was performed for all biological experiments. The collected data were analyzed using Microsoft Excel 2016 Analysis ToolPak. First, the one-way analysis of variance (ANOVA, San Francisco, CA, USA) test was performed to reveal if there were statistically significant differences between the tested groups (*p* value < 0.05 means that there were statistically significant differences in the studied groups). Tukey’s post-hoc test was used to calculate HSD (honestly significant difference) at the 5% level of significance. Different lowercase letters indicate significant differences between treatments.

## 4. Conclusions

In this study, on the basis of a glycine structure as a valuable building block and versatile initial component, a novel homologous series of ionic liquids (ILs) were successfully synthesized via an environmentally friendly approach and subsequently characterized. The anion source in the obtained products was 2,4-dichlorophenoxyacetic acid (2,4-DP), which is still being utilized in various commercial herbicide formulations worldwide. An appropriately designed synthesis path allowed various principles of green chemistry, such as *High Atom Economy*, *Less Hazardous Chemical Syntheses*, *Designing Safer Chemicals* as well as *Safer Solvents and Auxiliaries*, to be applied. Analysis of products’ physicochemical properties revealed insightful and valuable information regarding their interactions at the molecular level. The viscosity results, higher than those commonly found in the literature characteristic for ILs, can be explained by the presence of an ionic bond in the cation and carboxyl group, and hence forming intermolecular hydrogen bonds. The ILs’ densities and refractive indices linearly decreased with an increase in temperature, whereas the viscosity reduction was characterized by an exponential trend. The greenhouse test showed the high effectiveness of ILs in regulating the growth of oilseed rape (*Brassica napus* L.). The majority of them surpassed not only a salt of 2,4-DP comprising inorganic potassium cation, but also a reference preparation currently available on the market. The most effective compound possessed a dodecyl alkyl substituent, suggesting that the optimal lipophilicity could improve its adsorption into plant’s tissues more quickly, thus improving herbicidal activity. The presented results indicate that appropriately designed and obtained ILs have potential to become the new generation of plant protection products, complying with the principles of both green chemistry and sustainable agriculture.

## 5. Patents

Work reported in this manuscript is partially associated with the Polish patent: “Alkyldimethyl(carboxymethyl)ammonium 2-(2,4-dichlorophenoxy)propionates, method for obtaining them and application as a herbicides” (PL 230983).

## Data Availability

The data that support the findings in the present study are available from the corresponding author upon request.

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
