# Peer review of "Adjustment of the Structure of the Simplest Amino Acid Present in Nature—Glycine, toward More Environmentally Friendly Ionic Forms of Phenoxypropionate-Based Herbicides"

_ijms, 2023, doi:10.3390/ijms24021360_

Round 1

Reviewer 1 Report

The manuscript describes the series of novel herbicides, containing the glycine moiety in the cation together with the complex evaluation of their properties to provide comprehensive structure-property relationships. The study is very well designed, structured and clearly presented as well as it brings a significant novelty in the context of bio-inspired molecular design of herbicides, therefore, I recommend it for publication in International Journal of Molecular Sciences.

Few minor suggestions are as follows:

The authors show in the introduction that glycine is a valuable building block for the cations of the ILs due to the nitrogen atom present in its structure. The authors might want to mention also possible use of the glycine as an anion of the IL to provide wider outlook on this versatile bio-derived molecule.

This study adheres well to the principles of green chemistry in term of chemical design. The designed synthetic path via quaternization, protonation and anion metathesis reactions should give high Green Chemistry Metrics based on the reported high yields for the intermediate as well as final products. I would recommend to calculate them to underline and highlight the great sustainability aspect of the reported novel chemicals.    

The study involves a detailed synthetic description; the authors might want to mention the yields calculated for specific compounds also in the experimental section (materials and methods) or SI for clarity, specify the meaning of b/c/cd/d notations in figure 5 description as well as add AAILs term into the keywords to increase the visibility of the article.

I would also recommend to reconstruct the statement in the conclusions on the synthesis of ILs comprising cations derived from the simplest amino acid – glycine since it might be misleading – the presented novel ionic liquids have got the motif of the glycine in their structures, yet they were not directly derived from the glycine amino acid.

Reviewer 2 Report

The paper describe the design, synthesis, and herbicidal properties of ionic liquids made of alkyl substituted glycinium cations and 2-(2,4-dichlorophenoxy)propanoate  anion, choosing procedures as green as possible.
All the experiments are clearly described and results are soundly discussed. Also, the herbicidal properties are promising.
Supplementary Information is useful and clear.
However, no assesment was made about toxicity against animals in general, nor humans in particular.
All considered, the paper deserves publication.
A minor but - to me - important point is the use of some chemical name and formula
I'd like more attention to chemical terms, that sometimes follow "use" (wrong) instead of "rule".
For example, "isopropanol" in Table 3 is a conceptually wrong name (although widely used, it is wrong). According to IUPAC, alcohols names are made adding the suffix "-ol" to the name of the parent alkane. As a matter of fact, "isopropane" does not exist. Therefore the correct  name is 2-propanol.
Also, It should be better (see, for example, scheme 1) to write ClCH2CH2CO2H, because -COOH, strictly speaking, implies O-O bond.
